American Society for Microbiology | Microbiology Spectrum

# Cross-Species Transmission of Bat Coronaviruses in the Americas: Contrasting Patterns between Alphacoronavirus and Betacoronavirus

Diego A. Caraballo[a,b]

[a]CONICET-Universidad de Buenos Aires, Instituto de Ecología, Genética y Evolución de Buenos Aires (IEGEBA), Ciudad Universitaria-Pabellón II, Buenos Aires, Argentina

[b]Universidad de Buenos Aires, Facultad de Ciencias Exactas y Naturales, Buenos Aires, Argentina

**ABSTRACT** Bats harbor the largest number of coronavirus (CoV) species among mammals, serving as major reservoirs of alphaCoVs and betaCoVs, which can jump between bat species or to different mammalian hosts, including humans. Bat-CoV diversity is correlated with host taxonomic diversity, with the highest number of CoV species found in areas with the highest levels of bat species richness. Although the Americas harbor a unique and distinctive CoV diversity, no cross-species transmission (CST) or phylogeographic analysis has yet been performed. This study analyzes a large sequence data set from across the Americas through a Bayesian framework to understand how codivergence and cross-species transmission have shaped long-term bat-CoV evolution and ultimately identify bat hosts and regions where the risk of CST is the highest. Substantial levels of CST were found only among alphaCoVs. In contrast, cospeciation prevailed along the evolution of betaCoVs. Brazil is the center of diversification for both alpha and betaCoVs, with the highest levels of bat species richness. The bat family Phyllostomidae has played a key role in the evolution of American bat-CoVs, supported by the highest values of host transition rates. Although the conclusions drawn from this study are supported by biological/ecological evidence, it is likely that novel lineages will be discovered, which could also reveal undetected CSTs given that sequences are available from 11 of the 35 countries encompassing the Americas. The findings of this study can be useful for conducting targeted discovery of bat-CoVs in the region, especially in countries of the Americas with no reported sequences.

**IMPORTANCE** Coronaviruses (CoVs) have a strong zoonotic potential due to their high rates of evolvability and their capacity for overcoming host-specific barriers. Bats harbor the largest number of CoV species among mammals, with the highest CoV diversity found in areas with the highest levels of bat species richness. Understanding their origin and patterns of cross-species transmission is crucial for pandemic preparedness. This study aims to understand how bat-CoVs diversify in the Americas, circulate among and transmit between bat families and genera, and ultimately identify bat hosts and regions where the risk of CoV spillover is the highest.

**KEYWORDS** bats, codivergence, coronavirus, cross-species transmission, host shift, phylogeny, spillover, virus

**B**ats (order Chiroptera) include more than 1,400 species representing 20% of mammal diversity (1). They are also among the most widespread and abundant mammals. Together with their unique capacity for flying, some of their characteristics, such as food choices, population structure, seasonal migration, and roosting behaviors, make them particularly suitable hosts and spreaders of viruses and other disease agents (2).

Address correspondence to dcaraballo@ege.fcen.uba.ar.

The authors declare no conflict of interest.

RNA viruses have high mutation rates that allow especially fast evolution and can produce sporadic human epidemics overcoming host-specific barriers, resulting in the following different outcomes: single spillover events such as rabies virus (RABV) infections in humans (3), transient endemic outbreaks such as Nipah virus (4), transient epidemics such as the severe acute respiratory syndrome coronavirus (SARS-CoV) (5), or pandemic outbreaks such as SARS-CoV-2 (6). Cross-species transmissions (CSTs) are relatively frequent events that depend only on prevalence and contact rates between zoonotic host and nonhost species. However, the vast majority of CSTs are single spillover events, which epidemiologically are dead ends. Long-term establishment (when the virus is maintained by a novel host species) are rare events and depend not only on prevalence and contact rates but also on the biological interaction (and evolvability) of the virus and the new host (7).

Coronaviruses (order, *Nidovirales*; family, *Coronaviridae*), single-stranded positive-sense RNA viruses with the largest nonsegmented RNA viral genomes (16 to 31 kb), are divided into two subfamilies, *Orthocoronavirinae* and *Letovirinae*. The former includes the genera *Alphacoronavirus* (alphaCoV); *Betacoronavirus* (betaCoV), coronaviruses related to Middle East respiratory syndrome CoV (MERS-CoV), SARS-CoV, and SARS-CoV-2; *Gammacoronavirus* (gammaCoV); and *Deltacoronavirus* (deltaCoV). Alpha and betaCoVs are found in mammals and likely originated in bats, while the latter two are found and originated in birds (8). Coronaviruses are able to jump cross-species barriers and rapidly adapt to new hosts, likely due to their large genome size, high recombination rates, and genomic plasticity (9).

Bats harbor the largest number of CoV species among mammals, with alphaCoVs more widespread and abundant than betaCoVs (10). Bat-CoV diversity is correlated with host taxonomic diversity, with the highest CoV diversity found in areas with the highest levels of bat species richness (11, 12). Bat-CoVs are able to jump to different mammalian hosts, including humans (8). The emergence of SARS-CoV-2 highlights the relevance of bat-CoVs to global health and why understanding their origin and patterns of CST is crucial for pandemic preparedness (13). In the Americas, bat-CoVs have been detected in 11 countries as of 2021 (14). Both alpha and betaCoVs have been reported, with a predominance of the former (15–20). Although the Americas harbor a unique and distinctive CoV diversity, no cross-species transmission or phylogeographic analysis has yet been performed.

This study analyzes all bat-CoV sequences from the Americas available up to 2021. A phylogeographic Bayesian statistical framework was applied to model virus transmission between bat host species and the spatial location of bat-CoVs over time. This study aims to understand how bat-CoVs diversify, circulate among, and transmit between bat families and genera and ultimately identify bat hosts and regions where the risk of CoV CST is the highest.

## RESULTS

**American bat-CoVs in the context of global CoV diversity.** A total of 7 well-supported reciprocally monophyletic groups were retrieved for bat alphaCoVs in the Americas, all of them characterized for circulating in one or two main host species (Fig. 1). These clades are interspersed with other global alphaCoV lineages. Except for the sequence isolated from *Eumops glaucinus*, the American bat betaCoV lineage is monophyletic, supporting a common origin for this group.

**Alpha coronaviruses.** The American bat alphaCoVs were divided into seven well-supported clades, each of them defined by their main hosts as follows: *Molossus* (A), *Myotis* (B, C), and several Phyllostomidae genera (D to G) (*Anoura, Artibeus, Carollia, Desmodus, Glossophaga, Mesophylla, Phyllostomus,* and *Sturnira*). Two types of CSTs can be identified as follows: (i) long-term CSTs, which can be identified as monophyletic groups composed of a novel host; and (ii) dead ends, which are typically single sequences representing transmission events that did not produce long-term establishment in a novel host. Within clade G, involving Phyllostomidae, the following long-term CSTs can be identified: *Artibeus→Carollia, Artibeus→Phyllostomus, Phyllostomus→Desmodus,*

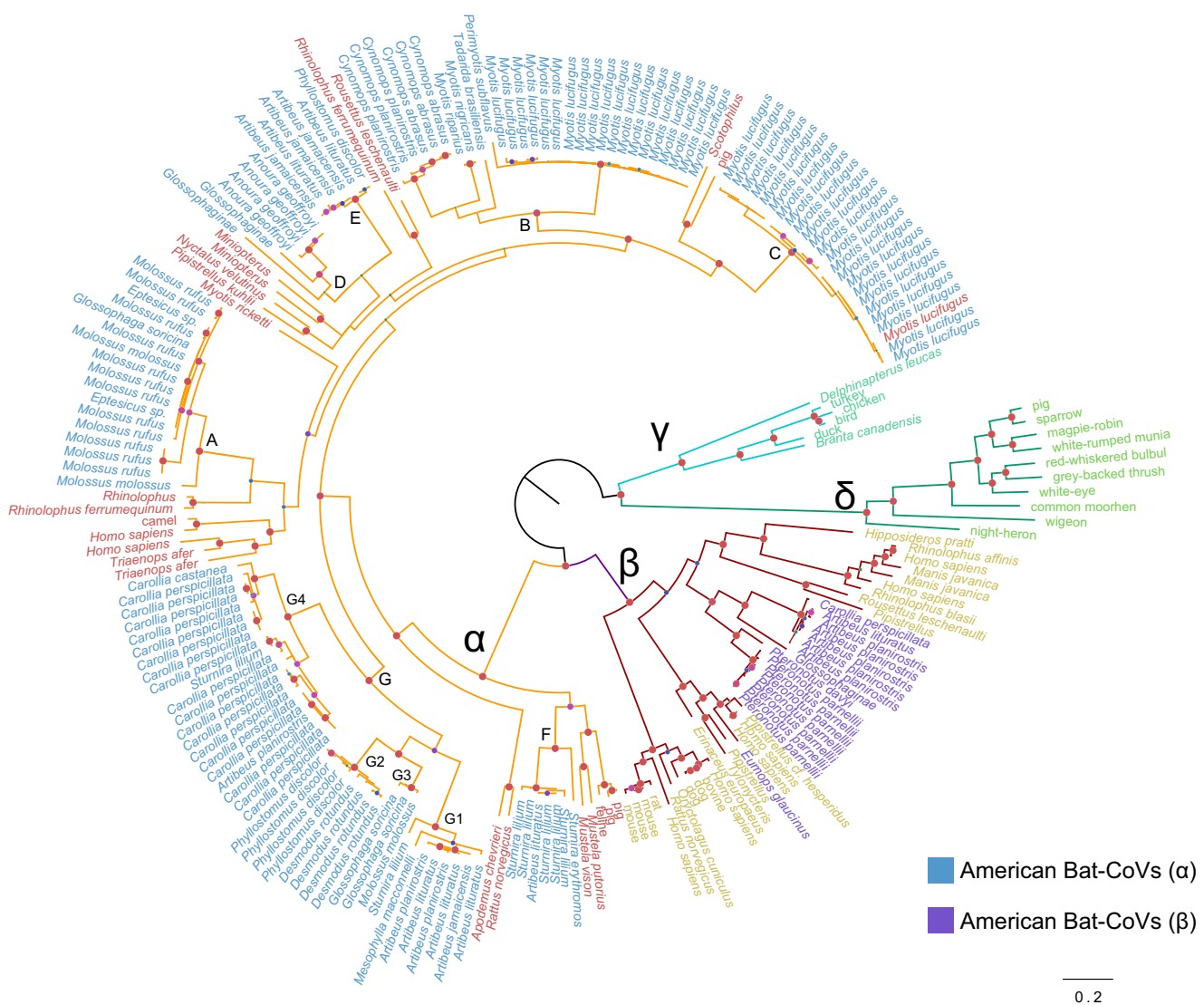

**FIG 1** American bat-CoVs in the context of global CoV diversity. Bayesian phylogeny showing global alpha, beta, delta, and gammacoronavirus. Sequences of American bat-CoVs are shown in blue (alphaCoVs) and purple (betaCoVs). Seven monophyletic American bat alphaCoVs are shown with letters A to G, and subclades of the major lineage G are shown with numbers 1 to 4. Nodes with Bayesian posterior probability >0.9 are shown in red. The scale bar is expressed in substitutions/site.

*Artibeus→Glossophaga* (Fig. 2). Another long-term CST can be identified from *Myotis* to *Cynomops* (clade B), which is also supported by the geographic location of the isolates since both donor and recipient sequences are from Brazil (see Supplementary Figure 1 and Supplementary Table 1 posted at https://doi.org/10.6084/m9.figshare.c.6029390.v1).

The majority of CSTs were observed in terminal branches, which likely correspond to dead ends in the transmission chain. All of these CSTs took place between species of the same (or neighboring) countries (Fig. 1; see also Supplementary Figure 1 posted at https://doi.org/10.6084/m9.figshare.c.6029390.v1), an expected result since a spillover would require physical contact. In clade A, *Molossus→Eptesicus* and *Molossus→Glossophaga* CSTs took place. It is worth noting that species from different families are involved in these events, the donor being Molossidae, while the recipients being Vespertilionidae and Phyllostomidae. These CSTs occurred between Brazilian bats. In clade B, *Myotis→Perimyotis* and *Myotis→Tadarida* CSTs took place. The donor sequences are from Canada, while the recipient sequences are from the United States.

A number of CSTs were also found among Phyllostomidae. The *Glossophaga→Anoura* CST (clade D) is plausible since there are sequences of both genera from the same region

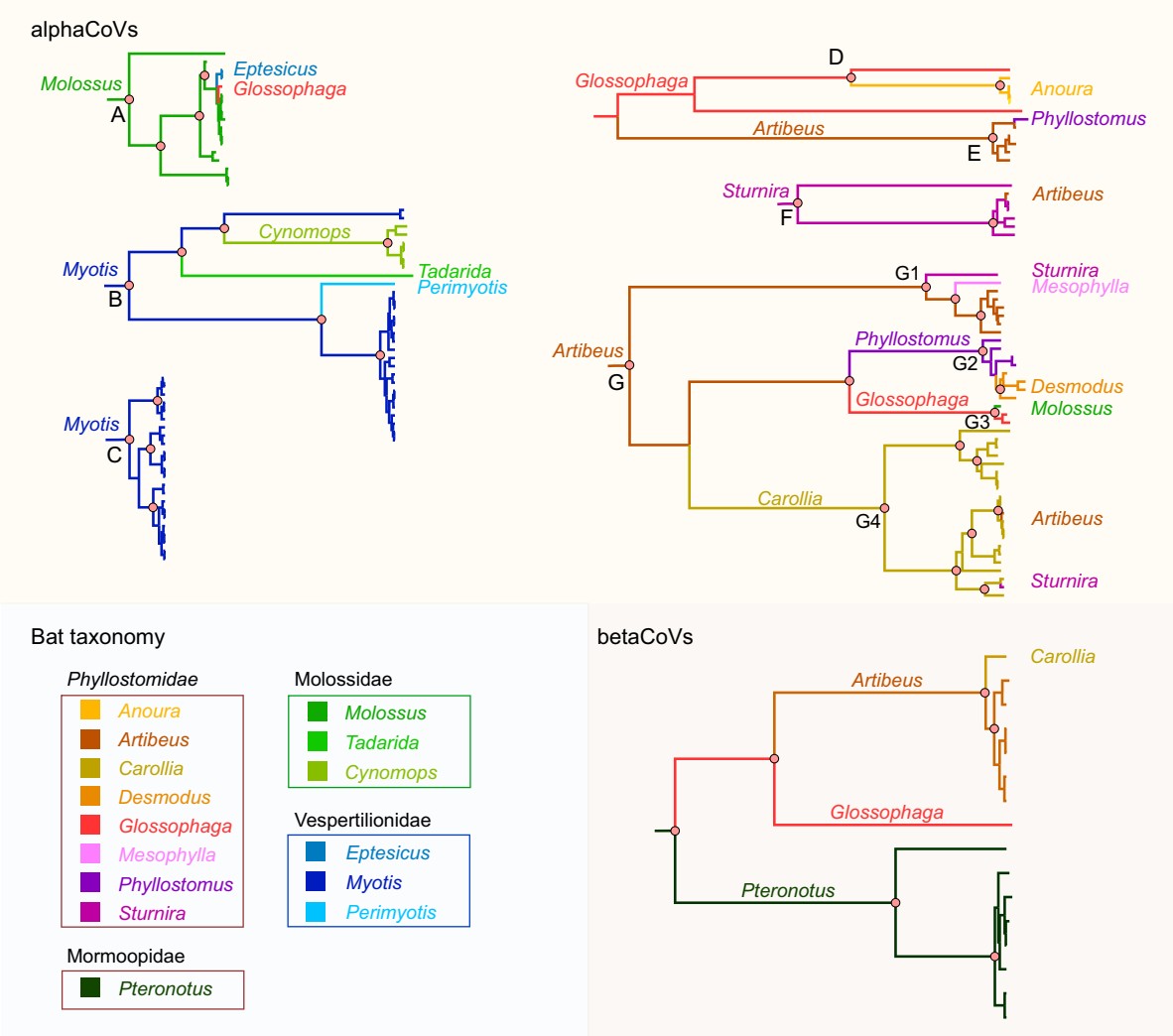

**FIG 2** Ancestral host reconstruction of alphaCoVs and betaCoVs. Maximum clade credibility trees annotated using partial RdRp sequences and bat host genus as discrete character state. Branch colors correspond to the inferred ancestral bat genus with the highest posterior probability. Relevant nodes with posterior probability >0.9 are indicated with pink circles.

(Costa Rica). In clade E, an *Artibeus→Phyllostomus* CST was identified. It is interesting to note that in this clade, there are sequences from Brazil, Panama, and Costa Rica, but the *Artibeus* and *Phyllostomus* sequences involved in the CST are both from Brazil. In clade F, *Sturnira* sequences are split into two groups reflecting geographic divergence. The *Sturnira→Artibeus* CST found in this clade occurred between sequences isolated from Brazilian bats. In clade G1, *Artibeus→Mesophylla* and *Artibeus→Sturnira* CSTs were found, supported by the isolation of variants from members of the three genera in Brazil. In clade G3, a *Glossophaga→Molossus* CST was found, involving Brazilian bats, and was the only interfamilial spillover found in the Phyllostomidae clades. Two additional CSTs were found in clade G4, *Carollia→Sturnira* and *Carollia→Artibeus*. All *Carollia* sequences split into two groups, one from Costa Rica, and the other from Brazil and Bolivia, reflecting the geographic structure of bat alphaCoVs. The CST events occurred between bats from Brazil (Fig. 2).

The cophylogeny shows both deep and terminal CSTs (Fig. 3). AlphaCoV clades A and B are, in general, congruent with Molossidae/Vespertilionidae phylogenetic relationships. Clades D to G include exclusively (except for terminal CSTs) variants found among Vespertilionidae. The fact that many species (genera *Artibeus*, *Desmodus*, *Sturnira*, and

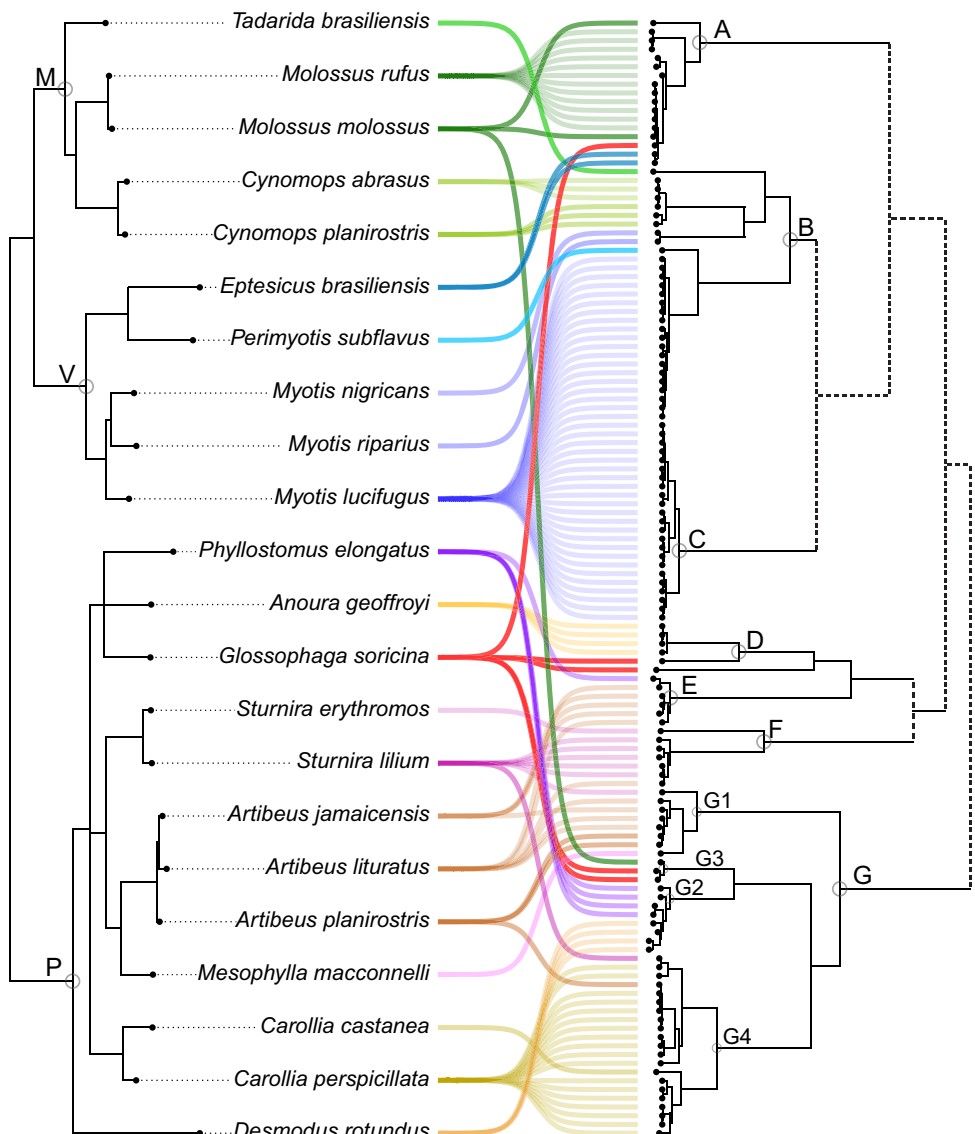

**FIG 3** Cophylogeny plot of bat species phylogeny and alphaCoVs tree. Cyt *b*-based bat species phylogeny (left) and alphaCoVs phylogeny (right). Gray circles indicate bat families (M, Molossidae; V, Vespertilionidae; P, Phyllostomidae) and main alphaCoV lineages and sublineages (A to G). Bat-CoV links are colored by each bat family. Dotted lines in the alphaCoV phylogeny depict nonsupported clusterings.

*Glossophaga*) have alphaCoVs from more than one clade (E, F, and G) indicates that these species maintain different alphaCoV variants.

The Bayes factor (BF) test enabled the distinction of three host-switching rate categories (Fig. 4). The transitions *Glossophaga→Molossus, Artibeus→Sturnira, Artibeus→Phyllostomus, Carollia→Artibeus,* and *Molossus→Eptesicus* received the highest support (BF > 100), all of them reflecting terminal CSTs. *Glossophaga→Molossus* is the host transition with the highest BF support, which is an expected result since CSTs in both directions were found (Fig. 2 and 3). The deep CST *Phyllostomus→Desmodus* also received high support (BF > 100).

There are six additional host switching rates that received strong BF support (10 < BF < 100), *Myotis→Perimyotis, Carollia→Sturnira, Artibeus→Mesophylla, Myotis→Tadarida, Myotis→Cynomps,* and *Glossophaga→Anoura.* Except for the latter two, which correspond to deep CSTs, the remaining are terminal CSTs. One last transition rate, with moderate BF support (3 < BF < 10), *Artibeus→Glossophaga,* can be considered. It is notable that *Artibeus* and *Molossus* function as both donor and receptor species in terminal CSTs.

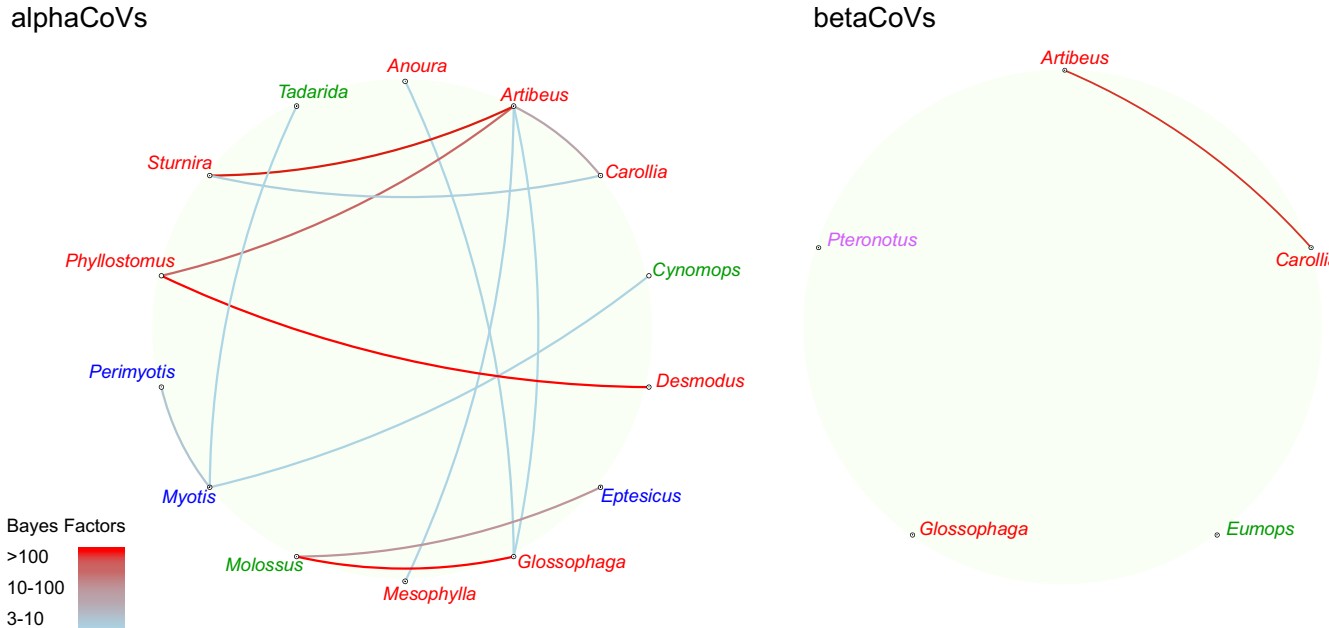

**FIG 4** Host transition rates among American bat genera. Strongly supported host switches between bat genera for alphaCoVs (left) and betaCoVs (right). Line color indicates the switch significance level according to the Bayes factor test. Genus color indicates host family (green, Molossidae; red, Phyllostomidae; blue, Vespertilionidae; purple, Mormoopidae).

**Betacoronaviruses.** The phylogenetic structure of the betaCoV tree shows a general correspondence with the host species phylogeny (Fig. 2). It can be divided into two well-supported clades defined by different host families, Phyllostomidae and Mormoopidae, also reflecting internal phylogenetic relationships. There are no observed terminal CSTs. The cophylogeny reflects a total correspondence between host and virus trees (Fig. 5), while only one host switching rate, *Artibeus→Carollia*, received strong BF support (10 < BF < 100), denoting low levels of CSTs found among betaCoVs (Fig. 3).

The cospeciation test yielded two maximum parsimony reconciliations (MPRs), one under each cost set (Fig. 6). Cost set *a* (2, 6, and 1; for duplications, transfers, and losses, respectively), invokes 5 cospeciation events, 10 duplications, and 1 loss. Cost set *b* (4, 2, and 1), hypothesizes 5 cospeciation events, 9 duplications, and 1 transfer. The *P* value of the randomization tests was <0.01 under cost set *a*, refuting the null hypothesis that host and virus trees are similar by chance. The MPR of condition *b* was not statistically different from randomized associations. These results indicate that the diversification of betaCoVs and American bats can be fully explained by cospeciation (and duplication/loss) events in the absence of host shifts.

## DISCUSSION

This study presents the first phylogenetic and phylodynamic analysis of bat-CoVs at a continental scale in the Americas. High levels of CoV diversity were found, which is expected, given that 75% of bat living genera are found in this region (1) and that bat-CoV diversity correlates with host taxonomic diversity. This is further supported by the fact that Brazil harbors the richest diversity of bat-CoVs found to date in the region (14; this study) as well as the highest levels of bat species richness at a global scale (21) (Fig. 7).

The results presented in this study confirm that the long-term evolution of bats and CoVs result from the interaction between codivergence and cross-species transmission, consistent with previous studies of bat-CoVs in other regions of the World (22–24). The phylogenetic analysis shows that, in both alpha and betaCoVs, there is at least some degree of species- and genus-specific tropism (25). However, both deep and shallow

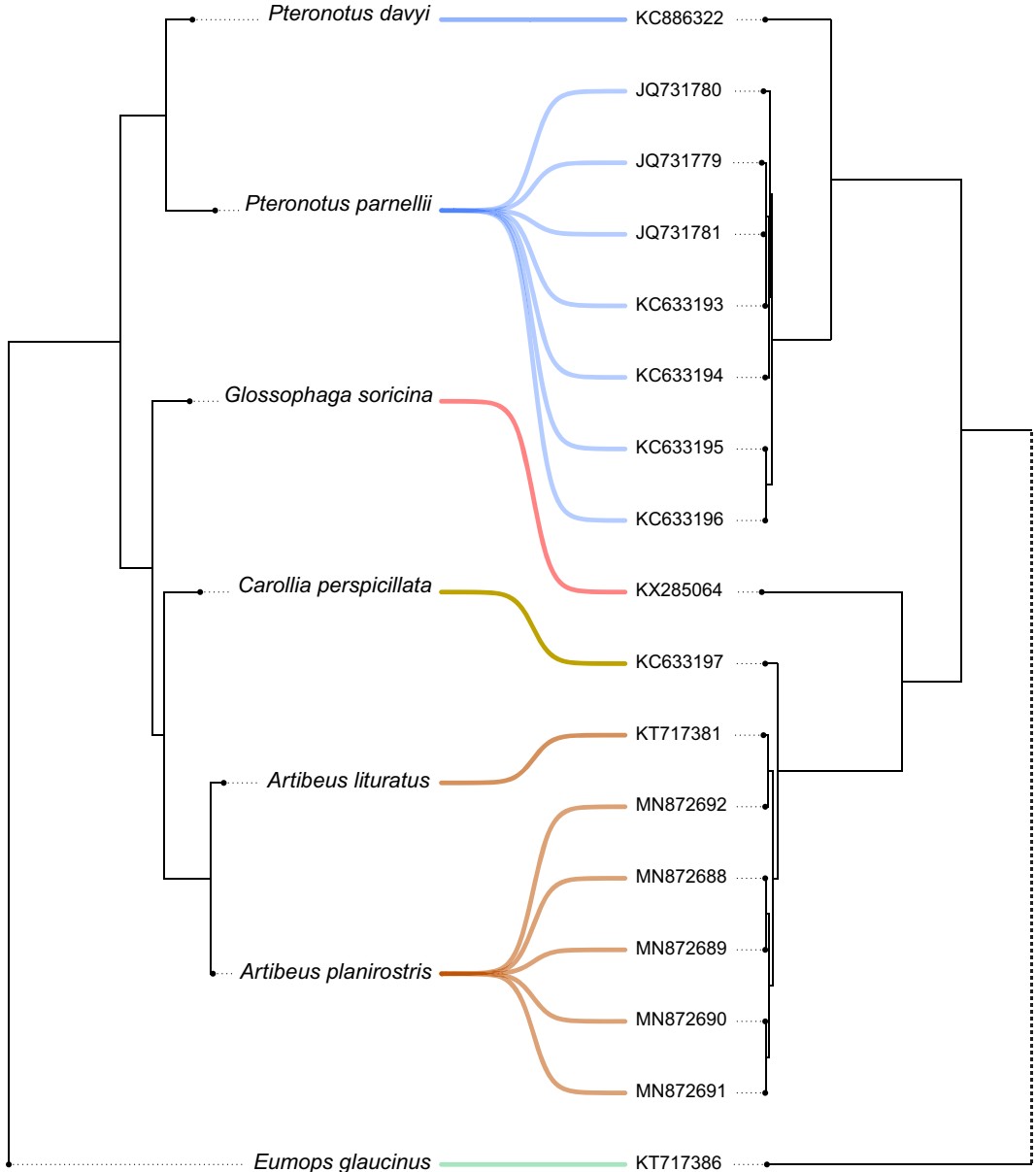

**FIG 5** Cophylogeny plot of bat species phylogeny and betaCoVs tree. Cyt *b*-based bat species phylogeny (left) and alphaCoVs phylogeny (right). Bat-CoV links are colored by each bat family (blue, Mormoopidae; green, Molossidae; red-yellow-brown, Phyllostomidae). Dotted lines in the betaCoV phylogeny depict nonsupported clusterings.

CSTs prevailed among alphaCoVs, as evidenced by the lack of monophyly of these sequences within the global CoV tree (Fig. 1) and by a high number of terminal CSTs between nonrelated species (Fig. 2). In contrast, the existence of genetically related viruses found among related bat species (or conspecifics) in distant sampling locations reflects codivergence between CoVs and bat species. This is the case of *Myotis*, *Sturnira*, *Phyllostomus*, *Carollia*, and *Artibeus* alphaCoVs and *Pteronotus* betaCoVs (see Supplementary Figure 1 posted at https://doi.org/10.6084/m9.figshare.c.6029390.v1).

The *Betacoronavirus* cophylogeny is entirely congruent, showing that this virus genus is less prone to host switching compared to *Alphacoronavirus*, which is in agreement with previous findings (23). The diversification of betaCoVs accompanied the diversification of host species, at the family, genus, and species levels (Fig. 2 and 5). Only cospeciation and duplication/loss events were recovered in the MPR, underlying the prevailing role of codiversification and discarding any CST in this viral lineage.

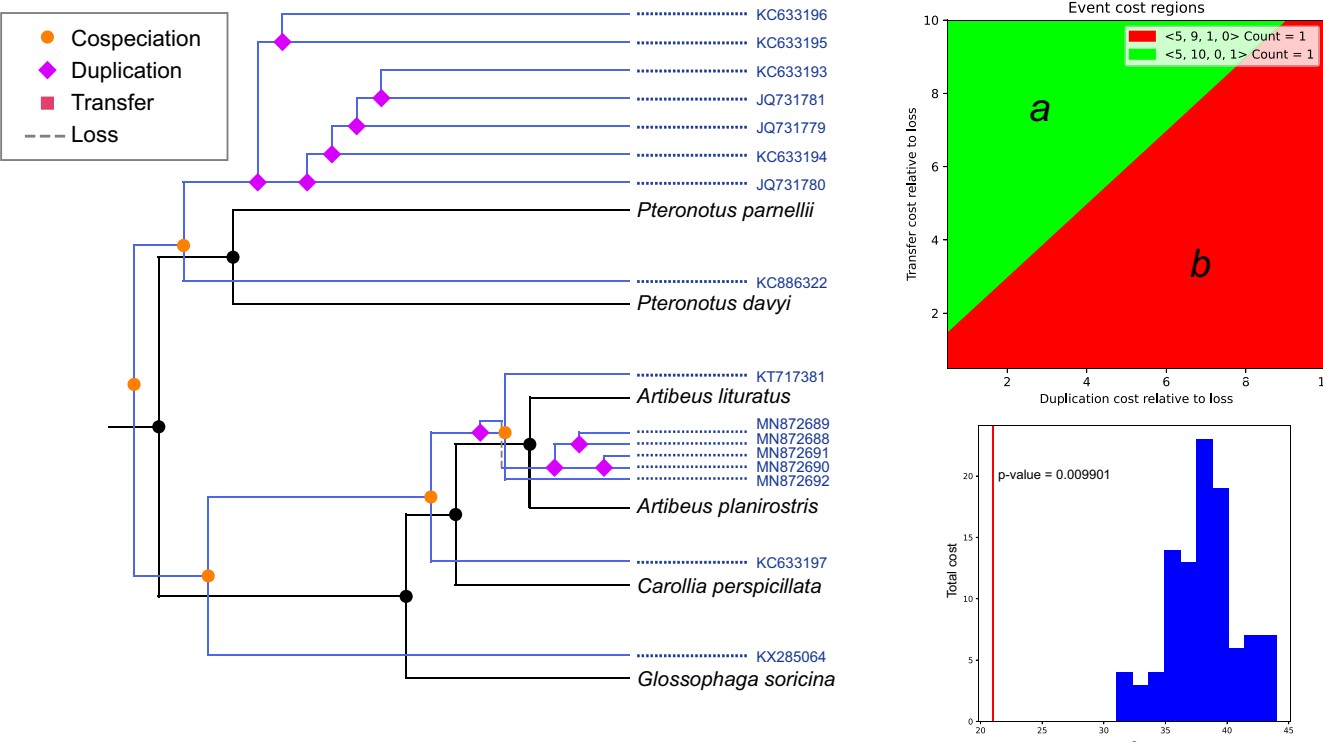

**FIG 6** Phylogenetic reconciliation between betaCoVs and bat hosts trees. Maximum parsimony reconciliation (left) proposed for the cost region *a* (upper right) and its significance by randomization tests (lower left). Cospeciations and duplications/losses but not host switching took place in the evolution of betaCoVs. The MPR obtained under scheme *b* (not shown) was not significantly different from the randomized MPR distribution.

It is interesting to note that inferred CSTs took place between species distributed in the same region. Deep CSTs occurring between bats from different families are unambiguous host shifts. Such host shifts were found in alphaCoVs between *Myotis* (Vespertilionidae) and *Cynomops* (and probably *Tadarida*) (Molossidae) (Fig. 2). In addition, several host shifts have been identified among Phyllostomidae as mentioned above (Fig. 2). Terminal CSTs took place both between closely related host species and between members of different bat families (Fig. 2). At least 10 terminal CSTs were detected among 128 alphaCoV sequences, denoting the high promiscuity of this virus genus in bats. These high CST rates were statistically corroborated; all bat genera included in the analysis depicted significant host-switching rates (Fig. 4). Except for *Glossophaga→Molossus*, all highly supported host switching rates were found between members of Phyllostomidae (Fig. 4), indicating that this family played a key role in the evolution, diversification, and cross-species transmission history of alphaCoVs in the Americas. It is notable that members of this family, such as *Carollia*, *Phyllostomus*, and *Glossophaga*, have been shown to share mixed-species roosts between them and with other species (26). The propensity of interspecific communal roosting has been attributed to benefits, such as protection against predation, information transfer on the location of foraging sites, and reduced thermoregulatory costs, but these benefits have the counterpart of higher chances for CST (27; this study).

Despite its large scale, this study has some limitations that should be considered. First, it is based on partial RNA-dependent RNA polymerase (RdRp) sequences since the majority of bat-CoV sequences available in public databases were generated with the primers designed by Watanabe et al. (28), which produce a fragment of 440 bp. The RdRp gene is a suitable marker for phylogenetic and phylogeographic analysis since it reflects vertical ancestry and is less prone to recombination than other regions of the CoV genome (29). However, the short length of the sequences is less than half of the 816 bp necessary for RGU (RdRp-based grouping units) classification (30) and is detrimental to phylogenetic resolution. It is worth noting that the general structure of the viral phylogenies inferred in

**Bat species**

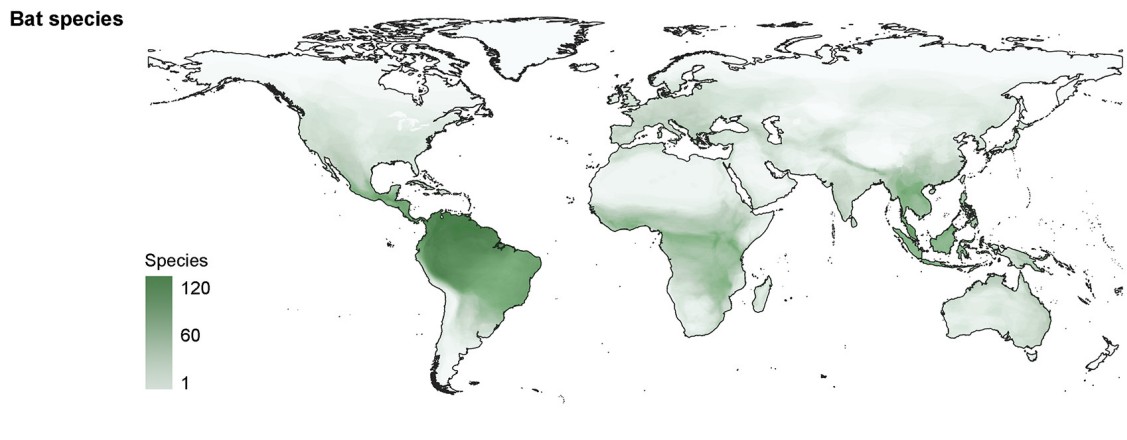

**alphaCoV host species**          **betaCoV host species**

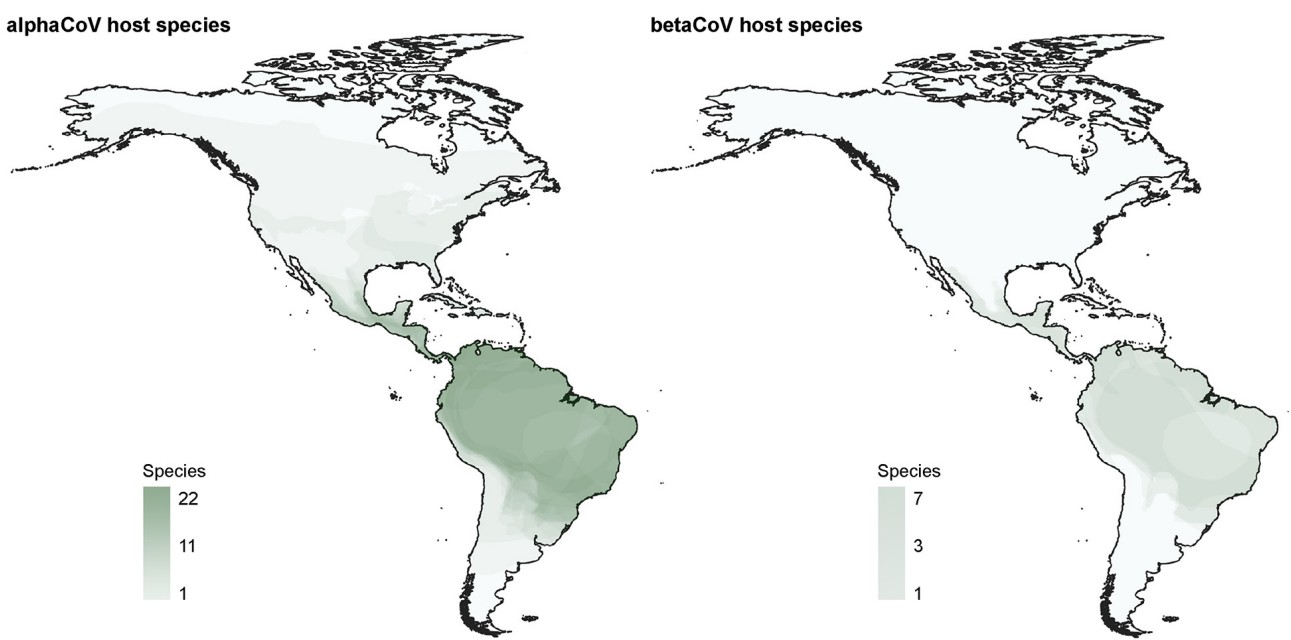

**FIG 7** Bat species distributions. Global bat species distributions (top) and distribution of bat-CoV hosts in the Americas (bottom). Geographical ranges of bat species were obtained from the International Union for Conservation of Nature (IUCN). The plots display the number of bat species based on overlapping geographical ranges from a total of 1,317 species with IUCN range data as of March 2022.

this study showed high levels of resolution with highly supported nodes. Another aspect that should be taken into consideration is the uncertainty in host species classification due to procedural errors or to the cryptic nature of several bat species that cannot be unambiguously identified from their external morphology (31). This issue was solved by centering the analysis on genus and family levels (Fig. 2, Fig. 4) but keeping the species-level resolution in the cophylogeny (Fig. 3, Fig. 5).

In this study, both host taxa and geographic regions that define the hot spot of CoV phylogenetic diversity in the Americas were identified. Namely, for both alpha and betaCoVs, Brazil is the center of diversification, with the highest number of lineages as well as the highest level of bat species richness (Fig. 7; see also Supplementary Figure 1 posted at https://doi.org/10.6084/m9.figshare.c.6029390.v1). Brazil has a humid tropical and subtropical climate, which has been demonstrated to promote higher viral evolutionary rates than other climates in other RNA viruses, such as rabies, but also in CoVs (23, 32). These regions are characterized by rapidly urbanizing human populations as well as high levels of livestock and poultry production, which, in addition to higher levels of CSTs and viral evolution, could favor disease emergence (33). Although

the conclusions drawn from this study are supported by biological/ecological evidence (e.g., CSTs occurred between species cooccurring in the same region, and higher rates were found between species sharing mixed-species roosts), this may not represent the whole picture since there are sequences available from 11 of the 35 countries encompassing the Americas. It is likely that novel lineages will be discovered, circulating in both previously identified as well as in novel hosts, which could also reveal undetected CSTs. Therefore, substantially wider sampling efforts will be required to encompass the complete diversity of CoVs in bats, especially in countries of the Americas that have not reported sequences yet.

As pointed out, Phyllostomidae has played a key role in the long-term evolution of bat-CoVs in the Americas, with the highest observed host transition rates. These findings can be useful for conducting targeted discovery of bat-CoVs in Brazil and neighboring countries as well as for the implementation of surveillance programs for early detection of outbreaks in people or livestock. In particular, the bat genera *Phyllostomus*, *Artibeus*, *Sturnira*, and *Glossophaga* deserve special attention in alphaCoV discovery. Coronavirus research in bats and other wildlife or domestic species will continue to be important for the identification of natural reservoirs and potential sources of spillover into the human population.

## MATERIALS AND METHODS

**Data acquisition and virus typing.** A total of 195 CoV nucleotide sequences were retrieved from the Virus Pathogen Database and Analysis Resource (ViPR) (34) through the website http://www.viprbrc.org/, filtering for *Orthocoronavirinae*, bats, and the Americas in the subfamily, host, and location fields, respectively. The resulting sequences covered the period of 2010 to 2019 and were mostly partial sequences encoding for the RNA-dependent RNA polymerase (RdRp) protein (see Supplementary Table 1 posted at https://doi .org/10.6084/m9.figshare.c.6029390.v1). All supplemental material is available at figshare (https://doi.org/10 .6084/m9.figshare.c.6029390.v1). The majority of the sequences were obtained with the primers designed by Watanabe et al. (28), frequently used in molecular diagnostic protocols. A total of 146 sequences encompassing this fragment were aligned (minimum length, 229 bp; maximum length, 2,779 bp; mean length, 462 bp) with Clustal Omega (35) and tested for phylogenetic affinity using reference sequences for alpha, beta, gamma, and delta CoVs representing the global CoV diversity based on a recently published data set (36). To this end, 1E8 Markov chain Monte Carlo (MCMC) generations were run in MrBayes v3.2.7 (37), under the GTR+I+G model selected with MrModeltest v2 for each codon position separately (38), discarding the first 25% of the run as burn-in. Trees were visualized in FigTree v1.4.4 (39). All tested sequences belonged to either alphaCoV (129 sequences) or betaCoV (17 sequences). The final data set included 145 sequences with their associated host species (Appendix; see also Supplementary Table 1 posted at https://doi.org/10.6084/ m9.figshare.c.6029390.v1). Host, country, and collection date metadata associated with each sequence was retrieved using the rentrez package (40), using the R environment (41) and RStudio (42).

**Phylogeographic and host-switching analysis.** The spatial and CST dynamics of alphaCoVs (128 sequences) and betaCoVs (17 sequences) were reconstructed using the software BEAST v1.10.4 (43). The American bat alphaCoV data set was divided into seven reciprocally monophyletic groups, while 16 of 17 American bat betaCoV sequences were monophyletic (Fig. 1). Each monophyletic group (except clades D and E, which comprise a single higher-order clade) was treated separately in the cross-species transmission analyses.

Sampling dates were used for molecular clock calibrations. Selected substitution models were HKY+I+G and HKY+I for alphaCoVs and betaCoVs, respectively. Base frequencies, substitution rate parameters, and rate heterogeneity models were unlinked across codon positions. A symmetric substitution model was selected for host and country ancestral state reconstruction, while a Bayesian stochastic search variable selection (BSSVS) procedure was chosen to identify hosts transition rates that adequately explain the phylogenetic diffusion process. A strict clock and a constant population size coalescent prior were used for these analyses (Appendix). Convergence and mixing of the MCMC were analyzed using Tracer v1.7.2 (44), combining two independent runs with LogCombiner v1.10.4. Trees were summarized with the maximum clade credibility (MCC) option, discarding 10% of the run as burn-in, using TreeAnnotator v1.10.4. Effective sample sizes (ESS) were >200 for all parameters.

The program SpreaD3 v0.9.7.1 (45) was used to visualize the output of the Bayesian phylogeographic and host-switching analyses. The MCC trees obtained were used to analyze host transition rates through a Bayes factor test.

**Cophylogeny.** Host phylogenies were obtained based on cytochrome *b* (Cyt *b*) nucleotide sequences retrieved from GenBank, one for alphaCoVs host species and another for betaCoVs host species (Appendix; see also Supplementary Table 2 posted at https://doi.org/10.6084/m9.figshare.c.6029390.v1). The cophylogenies of alphaCoVs, betaCoVs, and their respective hosts were obtained using the function cophylo included in the R package phytools v0.7-70 (46).

The American bat betaCoV tree was statistically tested for cospeciation using the software eMPRess (47), which implements the duplication-transfer-loss (DTL) model for maximum parsimony reconciliation (MPR). This software is advantageous as it allows setting the event cost landscape as a two-dimensional plot, and different regions can be established for cost combinations that yield the same MPRs. For each

cost, a variable number of MPRs can be obtained, which in turn can be clustered to minimize the distance between different sets of solutions. The betaCoV data set yielded two cost regions. Reconciliations were computed for both cost regions (cost set *a*, 2, 6, and 1; cost set *b*, 4, 2, and 1 [for duplications, transfers, and losses, respectively]). The resulting MPRs were statistically tested through a randomization test, consisting of 100 replicates permuting tip associations between hosts and viral sequences.

**Species richness maps.** Bat species richness maps were built using the software QGIS 3.16.0 Hannover (48), based on the IUCN Red List of Threatened Species (49) distribution shapes of all Chiroptera and selecting host species for the alphaCoVs and betaCoVs included in this study.

## ACKNOWLEDGMENTS

The author is grateful to María Gracia Gervasi whose comments and suggestions have improved this manuscript.

The Virus Pathogen Database and Analysis Resource (ViPR) (34) has been wholly funded with federal funds from the National Institute of Allergy and Infectious Diseases, National Institutes of Health, Department of Health and Human Services under contract number 75N93019C00076.

## APPENDIX

**Final CoV data set.** The final data set included 128 alphaCoV and 17 betaCoV sequences with their associated host species. The sequence MT734810 (alpha coronavirus) was removed from the analysis because it had an unspecified host. Two sequences were annotated as isolated from Glossophaginae species. These were treated as *Glossophaga soricina* since this was the only representative of Glossophaginae found in other sequences. Two sequences were annotated as isolated from *Eptesicus* species. These were treated as *Eptesicus brasiliensis* for being the most widespread and abundant species in Brazil, the country where these sequences were obtained (see Supplementary Table 1 posted at https://doi.org/10.6084/m9.figshare.c.6029390.v1).

**Molecular clock prior setting.** An initial analysis was run for 1E7 MCMC to check parameter mixing. The default clock prior was quite diffuse, so the posterior of this first run was used to set a more informative prior in subsequent analysis. In the final analysis, a normal distribution was chosen, with a mean of 3E−3 substitutions/site/year (s/s/y) and a standard deviation (SD) of 2E−4 s/s/y, truncated between 2.5 and 3.5 E−3 s/s/y.

**Cyt *b* data curation.** The Cyt *b* sequence of *Perimyotis subflavus* is annotated as *Pipistrellus subflavu*s (a synonym of the former). *Phyllostomus discolor* had partial sequences with several stop codons, probably being a pseudogene. The sequence of *Phyllostomus elongatus* was used to represent this genus.

**Host phylogenies.** The alphaCoV host tree included 22 bat species corresponding to the families Vespertilionidae, Molossidae, and Phyllostomidae. A total of 1E7 MCMC generations were run in MrBayes, using HKY+I+G and GTR+I+G for the first+second and the third codon positions, respectively, setting the burn-in to 2.34% of the run, according to the convergence criterion mentioned above.

The betaCoV host tree included seven species belonging to the families Mormoopidae, Phyllostomidae, and Molossidae (see Supplementary Table 2 posted at https://doi.org/10.6084/m9.figshare.c.6029390.v1). A total of 1E7 MCMC generations were run in MrBayes, using HKY+G and HKY+I+G for the first+second and the third codon positions, respectively, setting the burn-in to 3.8% of the run.

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
