## [Reviewer comments · Microbiology Spectrum]

Microbiology Spectrum

Cross-species transmission of bat coronaviruses in the Americas: contrasting patterns between alphacoronavirus and betacoronavirus.

Diego Caraballo

Corresponding Author(s): Diego Caraballo, Instituto de Ecología, Genética y Evolución de Buenos Aires

Review Timeline:

Submission Date:	April 22, 2022
Editorial Decision:	May 9, 2022
Revision Received:	June 3, 2022
Editorial Decision:	June 5, 2022
Revision Received:	June 6, 2022
Accepted:	June 6, 2022

Editor: Biao He

Reviewer(s): The reviewers have opted to remain anonymous.

Transaction Report:

DOI: <https://doi.org/10.1128/spectrum.01411-22>

May 9, 2022

Dr. Diego A. Caraballo
Instituto de Ecología, Genética y Evolución de Buenos Aires
Universidad de Buenos Aires, CONICET
Intendente Guiraldes 2160
Ciudad Universitaria, Pabellón 2, 4to piso
Buenos Aires 1428
Argentina

Re: Spectrum01411-22 (Cross-species transmission of bat coronaviruses in the Americas: contrasting patterns between alphacoronavirus and betacoronavirus.)

Dear Dr. Diego A. Caraballo:

Thank you for submitting your manuscript to Microbiology Spectrum. Generally, the topic is interesting, but the sequence dataset is very limited, leading to the conclusions are not well supported. I would like to expect a significant improvement of this study.

Link Not Available

Sincerely,

Biao He

Journals Department
Reviewer comments:

Reviewer #1 (Comments for the Author):

In this manuscript, the authors proposed a routine phylogenetic analysis on less than 200 RdRp protein sequences of coronaviruses, trying to identify the cross-species transmissions of the virus among bats. Overall, the limited dataset and the rough data processing can not support the conclusions in the MS.

1. The genome dataset is very limited, yet the author used just the RdRp gene sequences to interpret the possible transmission

events of coronaviruses, which was not a good choice. My suggestion would be using the genome data to analyze the recombinations of coronaviruses in the Americas, which would be more interesting.

2. Some sentences need rephrasing. For example, it is courageous to assert that "the evolution and diversification of these coronaviruses remains poorly studied".

Reviewer #2 (Comments for the Author):

In this manuscript, Caraballo assembled a large dataset of bat coronaviruses, employed phylogenetic approaches to analyze the evolution of alpha coronaviruses and beta coronaviruses in bats of the Americas, and identified hosts and regions with high spillover risk. The author reached several conclusions: (1) Cross species transmission were frequently identified in alpha coronaviruses; (2) Brazil appears to be the center of alpha and beta coronavirus diversification; (3) Phyllostomidae has the highest host transition rates. In my opinion, this manuscript provides some interesting insights into the evolution of coronaviruses of bats in the Americas, but the quality of writing should be improved. I have the following detailed comments:

Major comments:

- (1) The quality of writing should be improved throughout the manuscript. There are too many grammar errors, wording problems, and typos, and I give several examples here in several lines (L55 originate...epidemics, L57 CoV should be specified, L58 transmission are). Just too many to list!
- (2) The American CoVs should be put into the context of global CoV diversity. Therefore, a tree of global CoVs with American CoVs highlighted should be a nice add.
- (3) Recombination analyses should be performed before phylogenetic and molecular clock analyses.
- (4) Overall, I suggest the author not use cross species transmission, host shift, and spillover as slightly different concepts, which only makes the text difficult to understand. Cross species transmission is enough.
- (5) Once again, does the American betacoronaviruses form a monophyletic group? The cospeciation of beta CoVs should be tested statistically (Jane could be used).
- (6) Discussion is too long, and could be shorten.

Minor comments:

- (1) L57: I do not think SARS-CoV is a permanent epidemic.
- (2) L72-74: I do not think the current literature have established the relationship.
- (3) Figure 1: Posterior probability values should be labeled.
- (4) The 95% HPD for time should be included for each node.

Staff Comments:

Preparing Revision Guidelines

Please return the manuscript within 60 days; if you cannot complete the modification within this time period, please contact me. If you do not wish to modify the manuscript and prefer to submit it to another journal, please notify me of your decision immediately so that the manuscript may be formally withdrawn from consideration by Microbiology Spectrum.

Reviewer comments

Reviewer #1 (Comments for the Author):

In this manuscript, the authors proposed a routine phylogenetic analysis on less than 200 RdRp protein sequences of coronaviruses, trying to identify the cross-species transmissions of the virus among bats. Overall, the limited dataset and the rough data processing can not support the conclusions in the MS.

1. The genome dataset is very limited, yet the author used just the RdRp gene sequences to interpret the possible transmission events of coronaviruses, which was not a good choice. My suggestion would be using the genome data to analyze the recombinations of coronaviruses in the Americas, which would be more interesting.

Response:

The study was based on less than 200 RdRp sequences because there were no additional bat-CoV sequences for the region studied (the Americas). The procedure for selecting the final dataset was made by filtering for Bat coronavirus in the Americas through the VIPR website. This search yielded 195 sequences, most of which were partial RdRp sequences (of 400 bp) amplified with a widespread primer set (Watanabe et al. 2010). A fraction of these sequences were further discarded because they spanned a non-overlapping region, different from the rest of the sequences. The final dataset consisted of 145 overlapping bat-CoV sequences, maximizing the breadth of the analysis, which is centered on cross-species transmission. So, to have the highest host representation I decided to include the maximum number of overlapping sequences.

To illustrate the enormous disparity between full genomes and partial batCoV sequences, I would like to share with you the following figure, which summarizes the number of available batCoV sequences in the Americas.

The vast majority of available sequences are short (<1000 bp). There are only 2 full genomes which are insufficient for any host-shift analysis. To exemplify this situation, all the sequences from Brazil, the most probable hotspot for the diversification of bat-CoVs, are all shorter than 816 bp, and restricting the analysis to full genomes or multi-loci approaches would leave out the sequences of the most diverse region.

For the same reason, recombination tests cannot be performed with the actual dataset (I am convinced that recombination should be tested whenever it is possible. I underlined this necessity in a recent contribution on rabies virus: <https://dx.doi.org/10.3390/v13010023>). However, as I state in the manuscript, the RdRp gene is less prone to recombination than other regions of the CoV genome. In addition, even if they are the product of recombination, the finding of sequences that commonly circulate in one host species in a novel species, means that there was coinfection (and hence, cross-species transmission).

I agree that this dataset has inherent limitations, and as such, the conclusions that follow this analysis must be with precaution (and this is reflected in the Discussion section). But if a strict criterion of sequence length is applied, the analysis cannot be done, and the opportunity to start gaining insight into batCoVs at a continental scale for the Americas would be lost. I am sure there will be more batCoV sequences available in the next few years (I am personally working on generating full genome sequences in Argentina), but I am also convinced that these results could serve as good working hypotheses to guide future research and to point out precisely how important it would be to count with genomic analyses of virus and hosts diversity in the Americas.

2. Some sentences need rephrasing. For example, it is courageous to assert that "the evolution and diversification of these coronaviruses remains poorly studied".

Response: This sentence is referred specifically to bat-CoVs in the Americas, which indeed remain poorly studied, as reflected by the limited number of complete genomes as well as by the fact that most countries have no reported sequences. It was modified to better reflect this situation:

"However, due to the limited number of complete genomes and the lack of reported sequences in most of the Americas' countries, the evolution and diversification of these CoVs is poorly understood".

Reviewer #2 (Comments for the Author):

In this manuscript, Caraballo assembled a large dataset of bat coronaviruses, employed phylogenetic approaches to analyze the evolution of alpha coronaviruses and beta coronaviruses in bats of the Americas, and identified hosts and regions with high spillover risk. The author reached several conclusions: (1) Cross species transmission were frequently identified in alpha coronaviruses; (2) Brazil appears to be the center of alpha and beta coronavirus diversification; (3) Phyllostomidae has the highest host transition rates. In my

opinion, this manuscript provides some interesting insights into the evolution of coronaviruses of bats in the Americas, but the quality of writing should be improved. I have the following detailed comments:

Major comments:

(1) The quality of writing should be improved throughout the manuscript. There are too many grammar errors, wording problems, and typos, and I give several examples here in several lines (L55 originate...epidemics, L57 CoV should be specified, L58 transmission are). Just too many to list!

Response: The writing was checked and improved.

(2) The American CoVs should be put into the context of global CoV diversity. Therefore, a tree of global CoVs with American CoVs highlighted should be a nice add.

Response: This suggestion was taken into consideration and the results changed, in part, the focus of the paper. Although main clades remained unchanged, the lack of monophyly of bat alphaCoVs from the Americas, implied performing the analysis on each clade separately. This did not affect most conclusions but emphasized the importance of host switching in this group.

(3) Recombination analyses should be performed before phylogenetic and molecular clock analyses.

Response: I agree, but unfortunately these cannot be carried out when working with short sequences (See response to Reviewer one).

(4) Overall, I suggest the author not use cross species transmission, host shift, and spillover as slightly different concepts, which only makes the text difficult to understand. Cross species transmission is enough.

Response: Thank you for your suggestion, it was taken into consideration and the manuscript was changed accordingly. Although I still indicated a distinction between deep and terminal CSTs, because in many cases the former represent long-term establishment in novel hosts, while the latter generally are dead ends in the transmission chain.

(5) Once again, does the American betacoronaviruses form a monophyletic group? The cospeciation of beta CoVs should be tested statistically (Jane could be used).

Response: Yes, except for the sequence isolated in *Eumops* (which was removed for phylodynamic and cospeciation analyses), American bat betaCoVs form a monophyletic group. The suggestion was taken into consideration and cospeciation was tested statistically with eMPress (a newer version released by the developers of the software Jane).

(6) Discussion is too long, and could be shorten.

Response: The Discussion was shortened. In particular, the spatial diffusion reconstruction was removed from the analyses, on one hand, because it was improper for analyzing the

polyphyletic alphaCoVs, and on the other hand, because it did not contribute to enriching the CST analysis, which is the main focus of the study. Clade-specific ancestral location reconstructions were performed instead, and discussed in the manuscript (Supplementary Figure 1).

Minor comments:

(1) L57: I do not think SARS-CoV is a permanent epidemic.

Response: It was changed to *transient* epidemic.

(2) L72-74: I do not think the current literature have established the relationship.

Response: The sentence was rephrased, to suggest a probable link between CoVs' host-shifting ability and genomic/mutational properties of these viruses.

(3) Figure 1: Posterior probability values should be labeled.

Response: Showing numbers in multiple nodes would affect the readability of the figures. I opted to show them as colored circles, representing exclusively highly supported nodes (Bayesian Posterior Probability >0.9).

(4) The 95% HPD for time should be included for each node.

Response: The temporal analysis was removed, for the same reasons explained for the spatial diffusion analysis.

June 5, 2022

Dr. Diego A. Caraballo
Instituto de Ecología, Genética y Evolución de Buenos Aires
Universidad de Buenos Aires, CONICET
Intendente Guiraldes 2160
Ciudad Universitaria, Pabellón 2, 4to piso
Buenos Aires 1428
Argentina

Re: Spectrum01411-22R1 (Cross-species transmission of bat coronaviruses in the Americas: contrasting patterns between alphacoronavirus and betacoronavirus.)

Dear Dr. Diego A. Caraballo:

Thank you for your revision. As stated in the manuscript, the data set used here originated from partial Americas, so some conclusions might be biased due to uneven sampling. An additional discussion about such limit is better included in Abstract and Discussion. So I would like to expect another modification.

Link Not Available

Sincerely,

Biao He

Journals Department
Reviewer comments:

Staff Comments:

Preparing Revision Guidelines

To submit your modified manuscript, log onto the eJP submission site at <https://spectrum.msubmit.net/cgi-bin/main.plex>. Go to

Author Tasks and click the appropriate manuscript title to begin the revision process. The information that you entered when you first submitted the paper will be displayed. Please update the information as necessary. Here are a few examples of required updates that authors must address:

Please return the manuscript within 60 days; if you cannot complete the modification within this time period, please contact me. If you do not wish to modify the manuscript and prefer to submit it to another journal, please notify me of your decision immediately so that the manuscript may be formally withdrawn from consideration by Microbiology Spectrum.

Editor comments

Thank you for your revision. As stated in the manuscript, the data set used here originated from partial Americas, so some conclusions might be biased due to uneven sampling. An additional discussion about such limit is better included in Abstract and Discussion. So I would like to expect another modification.

Response: Thank you for the suggestion. The Discussion and Abstract were modified to reflect the fragmentary nature of the available dataset and to underscore the importance of CoV surveillance and typing in those countries with no reported sequences.

Reviewer comments

Reviewer #1 (Comments for the Author):

In this manuscript, the authors proposed a routine phylogenetic analysis on less than 200 RdRp protein sequences of coronaviruses, trying to identify the cross-species transmissions of the virus among bats. Overall, the limited dataset and the rough data processing can not support the conclusions in the MS.

1. The genome dataset is very limited, yet the author used just the RdRp gene sequences to interpret the possible transmission events of coronaviruses, which was not a good choice. My suggestion would be using the genome data to analyze the recombinations of coronaviruses in the Americas, which would be more interesting.

Response:

The study was based on less than 200 RdRp sequences because there were no additional bat-CoV sequences for the region studied (the Americas). The procedure for selecting the final dataset was made by filtering for Bat coronavirus in the Americas through the VIPR website. This search yielded 195 sequences, most of which were partial RdRp sequences (of 400 bp) amplified with a widespread primer set (Watanabe et al. 2010). A fraction of these sequences were further discarded because they spanned a non-overlapping region, different from the rest of the sequences. The final dataset consisted of 145 overlapping bat-CoV sequences, maximizing the breadth of the analysis, which is centered on cross-species transmission. So, to have the highest host representation I decided to include the maximum number of overlapping sequences.

To illustrate the enormous disparity between full genomes and partial batCoV sequences, I would like to share with you the following figure, which summarizes the number of available batCoV sequences in the Americas.

The vast majority of available sequences are short (<1000 bp). There are only 2 full genomes which are insufficient for any host-shift analysis. To exemplify this situation, all the sequences from Brazil, the most probable hotspot for the diversification of bat-CoVs, are all shorter than 816 bp, and restricting the analysis to full genomes or multi-loci approaches would leave out the sequences of the most diverse region.

For the same reason, recombination tests cannot be performed with the actual dataset (I am convinced that recombination should be tested whenever it is possible. I underlined this necessity in a recent contribution on rabies virus: <https://dx.doi.org/10.3390/v13010023>). However, as I state in the manuscript, the RdRp gene is less prone to recombination than other regions of the CoV genome. In addition, even if they are the product of recombination, the finding of sequences that commonly circulate in one host species in a novel species, means that there was coinfection (and hence, cross-species transmission).

I agree that this dataset has inherent limitations, and as such, the conclusions that follow this analysis must be with precaution (and this is reflected in the Discussion section). But if a strict criterion of sequence length is applied, the analysis cannot be done, and the opportunity to start gaining insight into batCoVs at a continental scale for the Americas would be lost. I am sure there will be more batCoV sequences available in the next few years (I am personally working on generating full genome sequences in Argentina), but I am also convinced that these results could serve as good working hypotheses to guide future research and to point out precisely how important it would be to count with genomic analyses of virus and hosts diversity in the Americas.

2. Some sentences need rephrasing. For example, it is courageous to assert that "the evolution and diversification of these coronaviruses remains poorly studied".

Response: This sentence is referred specifically to bat-CoVs in the Americas, which indeed remain poorly studied, as reflected by the limited number of complete genomes as well as by the fact that most countries have no reported sequences. It was modified to better reflect this situation:

“However, due to the limited number of complete genomes and the lack of reported sequences in most of the Americas’ countries, the evolution and diversification of these CoVs is poorly understood”.

Reviewer #2 (Comments for the Author):

In this manuscript, Caraballo assembled a large dataset of bat coronaviruses, employed phylogenetic approaches to analyze the evolution of alpha coronaviruses and beta coronaviruses in bats of the Americas, and identified hosts and regions with high spillover risk. The author reached several conclusions: (1) Cross species transmission were frequently identified in alpha coronaviruses; (2) Brazil appears to be the center of alpha and beta coronavirus diversification; (3) Phyllostomidae has the highest host transition rates. In my opinion, this manuscript provides some interesting insights into the evolution of coronaviruses of bats in the Americas, but the quality of writing should be improved. I have the following detailed comments:

Major comments:

(1) The quality of writing should be improved throughout the manuscript. There are too many grammar errors, wording problems, and typos, and I give several examples here in several lines (L55 originate...epidemics, L57 CoV should be specified, L58 transmission are). Just too many to list!

Response: The writing was checked and improved.

(2) The American CoVs should be put into the context of global CoV diversity. Therefore, a tree of global CoVs with American CoVs highlighted should be a nice add.

Response: This suggestion was taken into consideration and the results changed, in part, the focus of the paper. Although main clades remained unchanged, the lack of monophyly of bat alphaCoVs from the Americas, implied performing the analysis on each clade separately. This did not affect most conclusions but emphasized the importance of host switching in this group.

(3) Recombination analyses should be performed before phylogenetic and molecular clock analyses.

Response: I agree, but unfortunately these cannot be carried out when working with short sequences (See response to Reviewer one).

(4) Overall, I suggest the author not use cross species transmission, host shift, and spillover as slightly different concepts, which only makes the text difficult to understand. Cross species transmission is enough.

Response: Thank you for your suggestion, it was taken into consideration and the manuscript was changed accordingly. Although I still indicated a distinction between deep and terminal CSTs, because in many cases the former represent long-term establishment in novel hosts, while the latter generally are dead ends in the transmission chain.

(5) Once again, does the American betacoronaviruses form a monophyletic group? The cospeciation of beta CoVs should be tested statistically (Jane could be used).

Response: Yes, except for the sequence isolated in *Eumops* (which was removed for phylodynamic and cospeciation analyses), American bat betaCoVs form a monophyletic group. The suggestion was taken into consideration and cospeciation was tested statistically with eMPress (a newer version released by the developers of the software Jane).

(6) Discussion is too long, and could be shorten.

Response: The Discussion was shortened. In particular, the spatial diffusion reconstruction was removed from the analyses, on one hand, because it was improper for analyzing the polyphyletic alphaCoVs, and on the other hand, because it did not contribute to enriching the CST analysis, which is the main focus of the study. Clade-specific ancestral location reconstructions were performed instead, and discussed in the manuscript (Supplementary Figure 1).

Minor comments:

(1) L57: I do not think SARS-CoV is a permanent epidemic.

Response: It was changed to *transient* epidemic.

(2) L72-74: I do not think the current literature have established the relationship.

Response: The sentence was rephrased, to suggest a probable link between CoVs' host-shifting ability and genomic/mutational properties of these viruses.

(3) Figure 1: Posterior probability values should be labeled.

Response: Showing numbers in multiple nodes would affect the readability of the figures. I opted to show them as colored circles, representing exclusively highly supported nodes (Bayesian Posterior Probability >0.9).

(4) The 95% HPD for time should be included for each node.

Response: The temporal analysis was removed, for the same reasons explained for the spatial diffusion analysis.

June 6, 2022

Dr. Diego A. Caraballo
Instituto de Ecología, Genética y Evolución de Buenos Aires
Universidad de Buenos Aires, CONICET
Intendente Guiraldes 2160
Ciudad Universitaria, Pabellón 2, 4to piso
Buenos Aires 1428
Argentina

Re: Spectrum01411-22R2 (Cross-species transmission of bat coronaviruses in the Americas: contrasting patterns between alphacoronavirus and betacoronavirus.)

Dear Dr. Diego A. Caraballo:

I am glad to inform you that your manuscript has been accepted, and I am forwarding it to the ASM Journals Department for publication. You will be notified when your proofs are ready to be viewed.

Sincerely,

Biao He
Editor, Microbiology Spectrum
